# A Particle Swarm Optimization Based Approach to Pre-tune Programmable Hyperspectral Sensors

**Bikram Pratap Banerjee** [1,2] and **Simit Raval** [2,*]

1   Agriculture Victoria, Grains Innovation Park, 110 Natimuk Road, Horsham, VIC 3400, Australia; bikram.banerjee@agriculture.vic.gov.au
2   School of Minerals and Energy Resources Engineering, University of New South Wales, Sydney, NSW 2052, Australia
*   Correspondence: simit@unsw.edu.au; Tel.: +61-(2)-9385-5005

**Abstract:** Identification of optimal spectral bands often involves collecting in-field spectral signatures followed by thorough analysis. Such rigorous field sampling exercises are tedious, cumbersome, and often impractical on challenging terrain, which is a limiting factor for programmable hyperspectral sensors mounted on unmanned aerial vehicles (UAV-hyperspectral systems), requiring a pre-selection of optimal bands when mapping new environments with new target classes with unknown spectra. An innovative workflow has been designed and implemented to simplify the process of in-field spectral sampling and its realtime analysis for the identification of optimal spectral wavelengths. The band selection optimization workflow involves particle swarm optimization with minimum estimated abundance covariance (PSO-MEAC) for the identification of a set of bands most appropriate for UAV-hyperspectral imaging, in a given environment. The criterion function, MEAC, greatly simplifies the in-field spectral data acquisition process by requiring a few target class signatures and not requiring extensive training samples for each class. The metaheuristic method was tested on an experimental site with diversity in vegetation species and communities. The optimal set of bands were found to suitably capture the spectral variations between target vegetation species and communities. The approach streamlines the pre-tuning of wavelengths in programmable hyperspectral sensors in mapping applications. This will additionally reduce the total flight time in UAV-hyperspectral imaging, as obtaining information for an optimal subset of wavelengths is more efficient, and requires less data storage and computational resources for post-processing the data.

**Keywords:** evolutionary computation; heuristic algorithms; machine learning; unmanned aerial vehicles (UAVs); vegetation mapping; upland swamps; mine environment

## 1. Introduction

Hyperspectral technology is a potential tool for the remote detection of targets and monitoring. A hyperspectral sensor measures electromagnetic radiation reflected from the target in a large number of spectral narrowbands. The inherent objective in target classification and assessment using hyperspectral data is to utilize its high spectral resolution [1]. However, the large dimensionality of hyperspectral data is often attributed to the Hughes phenomenon, the curse of dimensionality [2]. The problem is a combined consequence of the high correlations among the adjacent bands and the inability of the algorithm being applied to process the high-dimensional data. The problem is paramount in spectrally complex environments such as wetlands and swamps with many diverse species to be monitored [1,3,4]. While a common remote sensing data processing solution involves the application of dimensionality reduction techniques or the selection of suitable narrowbands in a post-acquisition step, a hardware-based solution involves the use of programmable hyperspectral sensors as a pre-acquisition step. Programmable hyperspectral sensors typically involve a snapshot-based scanning mechanism, unlike general point or line scanning-type systems, which are non-programable and acquire a continuous spectrum

over the operable wavelength region. Several such programmable hyperspectral sensors have been developed in recent times, which are increasingly being used in UAV-based remote sensing applications [5–7]. A hardware-based method, such as Fabry–Pérot interferometer (FPI) technology, acquires reflected electromagnetic radiation in pre-selected optimal narrowbands, and it is programmed by changing the air gap between the internal tuneable mirrors [8]. This method has the additional benefit of efficient mapping of the environment through the selection of only the spectral features of interest, which is particularly crucial in high-resolution mapping applications using unmanned aerial vehicles (UAVs), which have limited flight times. The technology is relatively new compared to the traditional pushboom type hyperspectral sensors, and existing works involving the FPI have used either (1) a set of bands for generating vegetation indices (VIs), herein referred to as *indices-based* criteria [7,9,10], or (2) set of bands identified through rigorous experimental testing, herein referred to as *knowledge-based* criteria [11,12] of narrowband selection. *Indices-based* criteria for band selection have the potential to assess the condition and/or estimate the yield of the vegetation [7,9]; however, they are not principally suited for multi-target classification, since the spectral variations of the target endmembers present within the scene are subjective. Furthermore, the efficacy of *indices-based* narrowband selection approach for vegetation quality or condition assessment is also subject to the characteristic reflectance of the target, and the traditional list of indices does not always ensure the best results for different vegetation communities or species. The *knowledge-based* approach requires a thorough understanding of the spectral variability among the targets present over the area, which is usually attained through intensive in-situ sampling and is not always realizable over difficult terrain or in scenarios requiring urgent mapping. Therefore, it is important to adopt a *data-driven* methodology for programmable hyperspectral sensors to estimate appropriate narrow bands for scene classification or assessment. Minet et al. [13] proposed an approach to adaptively maximize the contrast between the targets by employing a genetic algorithm (GA)-based optimization of the positions and linewidths of a limited number of filters in FPI for military applications. However, this method is unsuitable in thematic applications of remote sensing.

Different *data-driven* strategies have been proposed for the selection of optimal bands for traditional remote sensing applications. A method of sub-optimal search strategy utilizing constrained local extremes in a discrete binary space to select hyper-dimensional features was presented in [14]. Becker et al. [3] used a second-derivative approximation to identify the spectral location of inflection. A band selection method using the correlations among bands based on mutual information (MI) and deterministic annealing optimization was also employed [15]. Becker et al. [4] proposed a classification-based assessment for three optimal spectral band selection techniques (derivative, magnitude, fixed interval, and derivative histogram), using the spectral angle mapper (SAM) as a classifier. A GA-based wrapper method using a support vector machine (SVM) was proposed for the classification of hyperspectral images [16]. A double parallel feedforward neural network based on radial basis function was used for dimensionality reduction [17]. Principal component analysis for identifying optimal bands to discriminate wetland plant species was presented [1]. A semi-supervised band clustering approach for dimensionality reduction was developed [18]. A particle swarm optimization (PSO)-based dimensionality reduction approach to improving support vector machine (SVM)-based classification was suggested by [19]. Li et al. [20] and Pal et al. [21] presented a hybrid band selection strategy based on a GA-SVM wrapper to search optimal bands' subsets. A method of band selection based on spectral shape similarity analysis was put forward in [22]. Methods for nesting a traditional single loop of PSO or 1PSO inside an outer PSO loop, termed 2PSO, have been identified to improve the overall optimization performance in certain applications, at the expense of computational cost [23]. Su et al. [23] implemented 1PSO and 2PSO with minimum estimated abundance covariance (MEAC) [24], among other techniques, for the evaluation of optimal bands. Ghamisi et al. [25] presented a feature selection approach based on hybridization of a GA and PSO with an SVM classifier as a fitness function. Accuracies

achieved in an optimized band selection method are influenced by the characteristics of the input dataset, as the search strategy depends on the present classes and their spectral profiles. Therefore, these methods need to be tested on benchmark datasets, an equivalent comprehensive evaluation is reported in [23]. However, all these existing optimal band identification studies involving *data-driven* methods were used on traditional hyperspectral datasets after the acquisition, and are yet to be used with a hardware-based solution to pre-tune hyperspectral sensors to acquire the optimal bands.

In this study, for the first time, an in-field *data-driven* approach to pre-tune a snapshot-type UAV-hyperspectral sensor was devised for remote sensing applications. The method employs PSO, with minimum estimated abundance covariance (MEAC), similarly to [23] in a post-processing stage for waveband selection after hyperspectral dataset acquisition. The significant benefits are: (1) it is an efficient approach to identifying the optimal bands in-field before the survey; (2) it does not require a lot of spectral samples per class, which is particularly an issue over difficult terrain when trying to establish a spectral library; and (3) the system works perfectly when the number of observed samples is less than the total number of potential hyperspectral bands to select from, which is an important issue with other dimensionality reduction methods, such as principal component analysis (PCA). Programmable UAV-hyperspectral sensors have increasingly been used in applications such as environmental mapping, precision agriculture, phenotyping, and forestry [12,26,27]. Identification of optimal wavelengths remains crucial for mapping vegetation communities, phenotyping functional plant traits, and identifying vegetation under biotic or abiotic stress. Our method aims to resolve functional challenges by improving the capturing of the spectral representation of an environment through a UAV-hyperspectral survey.

The rest of the paper is arranged as follows. The Materials and Methods section describes the experimental framework. The theoretical background of the PSO-MEAC approach is described in relation to the elements of the proposed application. In the Results and Discussion section, we present the results of using the PSO-MEAC method for optimal band selection at the experimental site. In addition, the performance of the *data-driven* PSO-MEAC approach has been evaluated against the traditional *indices based* approach for feature selection and mapping. Finally, the concluding remarks are provided in the conclusion section.

## 2. Materials and Methods

This section details the study area, ground based hyperspectral sensing system, data processing for the hyperspectral data, workflow for identifying optimal bands in the field, and method for UAV-hyperspectral surveying and assessment.

### 2.1. The Area Used for the Experiment

The test site is an upland swamp area above an underground coal mine within the temperate highland peat swamp on sandstone (THPSS) in New South Wales, southwest of the city of Sydney, Australia (34°21′24.0″S, 150°51′51″E). The area is located in Wollongong. The focus was laid on spectrally diverse vegetation communities in critically endangered ecosystems distributed in the Blue Mountains, Lithgow, Southern Highlands, and Bombala regions in New South Wales, Australia [28]. The NSW National Parks and Wildlife Service (NPWS) classifies the upland swamps complexes into five major vegetation communities—Banksia Thicket, Cyperoid Heath, Fringing Eucalypt Woodland, Restioid Heath, and Sedgeland [29]. The site has occasional thick vegetation cover and steep gradients which are inaccessible.

### 2.2. Hyperspectral Set-Up for Ground Based Sampling

The spectra of the target classes in the environment were measured with the visible-infrared snapshot hyperspectral (FPI) sensor (Rikola, Senop Optronics, Kangasala, Finland) with a separate data acquisition computer. In this mode of operation, the sensor acquires the maximum number of wavelength bands possible—i.e., 380 bands at 1 nm spectral

steps between 500 and 880 nm. With a focal length of 9 mm and a field-of-view (FOV) of 36.5 × 36.5 degrees, the sensor acquires 1010 × 1010 spatial channels in the snapshot imaging mode. In contrast, in the standalone on-board UAV-based data acquisition mode the sensor records a set of 15 programmed wavelength bands in 1010 × 1010 pixel format, i.e., up to a total of 16 megapixels of storage per hypercube. The sensor also acquires solar irradiance measurements—it uses an irradiance sensor for radiometric calibration; and positional measurements using a global positioning system (GPS) for geometric corrections (Figure 1). All sensors were installed on a handheld mount for hyperspectral imaging. An Android mobile phone was also installed on the sensor mount and paired to the data acquisition computer with a video telemetry feed over a WiFi link to provide a realtime view of the scene, which was useful for bringing the target vegetation in focus before the collection of hyperspectral data (Figure 1a). Additionally, a realtime feed of goniometric measurements (roll and pitch) from the mobile phone's accelerometer was relayed to the screen of the data acquisition computer to monitor the planimetric setting of the captured hypercubes using the FPI sensor (Figure 1b).

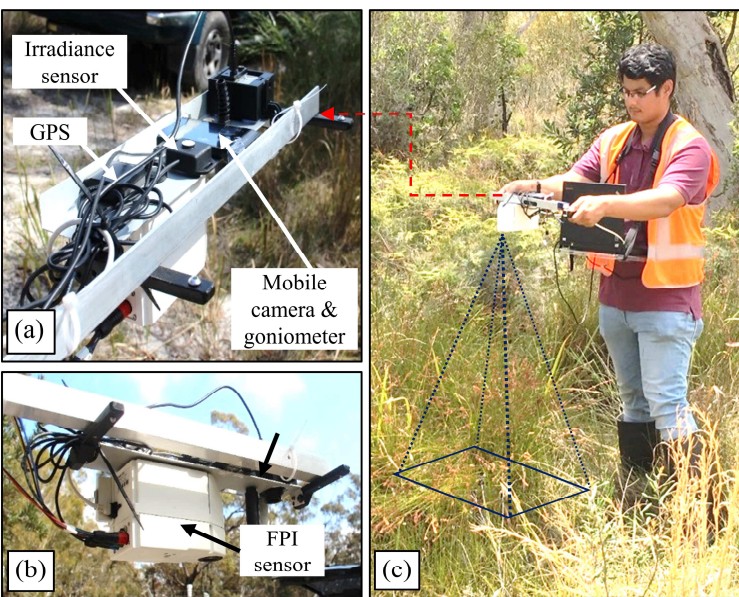

**Figure 1.** The setup for ground-based hyperspectral data acquisition using a Fabry–Pérot interferometer (FPI) sensor (Rikola, Senop Optronics, Kangasala, Finland), an irradiance sensor, a global position system (GPS), and an android phone as goniometer on a portable handheld sensor mount: (**a**) top-side view, (**b**) bottom-side view, and (**c**) in-field hyperspectral data acquisition with a data acquisition computer. The system was used for the collection of in-field data for rapidly identifying optimal hyperspectral wavelengths, for applications in aerial (UAV-hyperspectral) data acquisition.

The simplistic design of the handheld hyperspectral imaging system was important for carrying it around in regions with dense shrub-type vegetation cover (Figure 1c). The hyperspectral data were acquired with a downward nadir orientation over the shrub type swamp vegetation. The data were acquired at a distance of approximately 0.5 m from the top of the canopy (Figure 1c). In this study, the FPI sensor was used as a tool for in-field spectral acquisition to demonstrate an independent form of operation. Nevertheless, the field spectral measurements could also be obtained from other spectroradiometers, such as ASD FieldSpec3 (Analytical Spectral Devices, Boulder, CO, USA). However, special care should be taken to establish proper radiometric calibration to remove any inter-sensor response mismatch, which is addressed by using the same FPI sensor for both in-field spectral data collection for identifying the optimal bands and later UAV-hyperspectral data acquisition.

For identifying the optimal bands through PSO-MEAC, the hyperspectral measurements were collected for a total of three target vegetation classes, covering eight upland swamp species, including Grass tree (*Xanthorrhoea resinosa*), Pouched coral fern (*Gleichenia dicarpa*). and Sedgeland complex (*Empodisma minus, Gymnoschoenus sphaerocephalus, Lepidosperma limicola, Lepidosperma neesii, Leptocarpus tenax,* and *Schoenus brevifolius*). In addition, spectral measurements were also collected for background vegetation, which contained a mixture of other species which were present in small patches and not selected in this study. Finally, a background bare-earth spectrum was also collected. To obtain a proper un-mixed spectrum for a single species, field sampling was performed over a region of interest with local homogeneity.

### 2.3. In-Field Ground-Based Hyperspectral Data Processing

Vegetation in an upland swamp environment is highly diverse, and species can exist in homogenous and heterogeneous patches. Data collected through the portable handheld FPI system caused minor spectral misalignments due to unavoidable handheld movement of the sensor and due to slight movements of the canopy caused by wind. This happened as the data in the FPI sensor were acquired in a snapshot, bandwise manner with a small delay and sensor movement [26]. The hyperspectral bands were aligned using a previously developed band alignment workflow described in [26]. The data were first flat-field corrected using dark current removal and a white calibration panel; then they were converted to the reflectance measurements using previously computed calibration coefficients with an integrating sphere [7]. A band-averaged hyperspectral signal was calculated from the hypercube and used in the optimal band identification workflow. The spectrum was further treated using a Savizky–Golay [30] smoothing filter with a polynomial order of 3 and a frame length of 17 to remove spectral noise. A PSO with MEAC as the criterion function was employed to identify the suitable bands in the field; the details of the theory of operation are in Section 2.4. The entire process of spectral signature retrieval and PSO-MEAC workflow for suitable band identification was implemented as MATLAB (ver. 9.5) routines, and a graphical user interface (GUI) was designed for user-friendly and seamless operation in the field.

### 2.4. Optimal Band Identification Using PSO-MEAC

Particle swarm optimization (PSO) was originally used to simulate the social behaviour (movement and interaction) of the organisms (*particles*) in a flock of birds or a pool of fishes [31]. It has, however, been used as a robust metaheuristic computational method to improve the selection of candidate solutions for an optimization problem. The optimization operates iteratively over a swarm of candidate solutions with a criterion function as a given measure of quality. In our approach, the selected set of bands are called *particles*, and a recursive update of the bands is called a *velocity*. The particle position $x_{id}$ denotes the selected band subset of size $k$, and velocity $v_{id}$ denotes the update for the selected band. A particle updates [31] as shown in Equation (1).

$$v_{id} = \omega \times v_{id} + c_1 \times r_1 \times (p_{id} - x_{id}) + c_2 \times r_2 \times \left(p_{gd} - x_{id}\right)$$
$$x_{id} = x_{id} + v_{id}$$

(1)

where $p_{id}$ is the historically best local solution; $p_{gd}$ is historically the best global solution among all the particles; $c_1$ and $c_2$ control the contributions from local and global solutions, respectively; $r_1$ and $r_2$ are independent random variables between 0 and 1; and $\omega$ is the inertia weight to improve the convergence performance.

New velocities and positions ($v_{id}$ and $x_{id}$ on the left-hand side of Equation (1)) for the particles are updated based on the existing parameters and cost criterion upon every iteration (Figure 2). The iteration process aims to minimize the underlined criterion function.

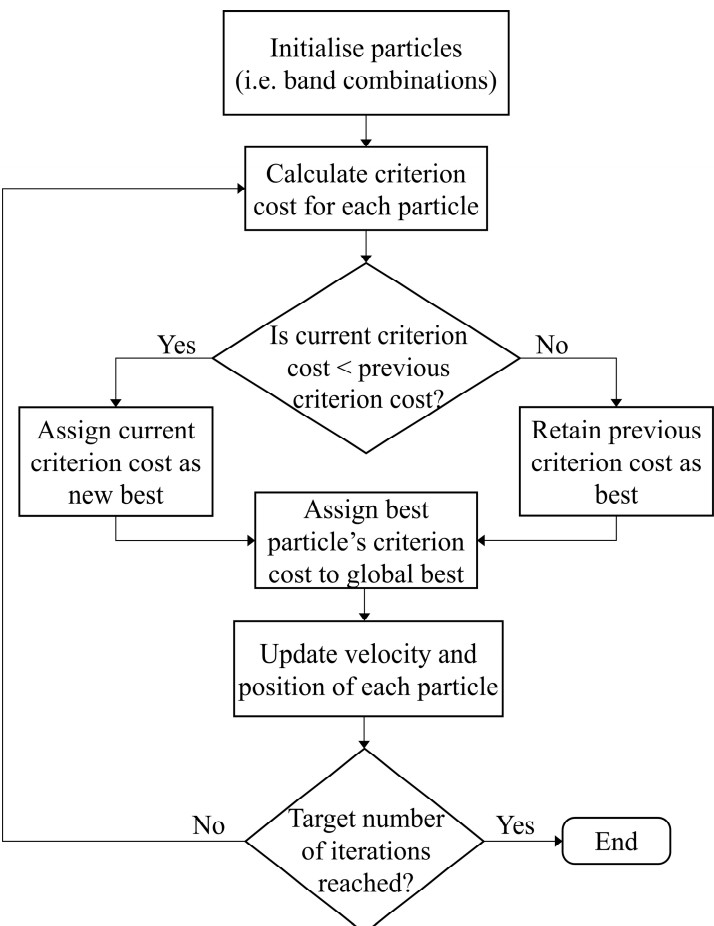

**Figure 2.** The method for the PSO-MEAC system. The algorithm initializes a set of particles or a combination of bands; at each iteration, the cost function (MEAC) associated with individual particles is computed; trajectories of the particles are projected towards the particle with the best solution; the loop is exited after the specified number of iterations is reached. The particle with a minimum cost function is identified as the optimal solution.

In a traditional supervised classification, where representative class signatures are known through exhaustive field surveying, the band-selection process can be greatly simplified. However, in an aerial survey to determine suitable wavelength bands for a programmable UAV-hyperspectral system, such an exhaustive exercise is tedious, cumbersome, and not always possible. Therefore, MEAC was used as a criterion function in PSO, as it requires only class signatures and no training samples. The efficacy of this technique has been previously evaluated against other existing optimization methods by Su et al. [23] for feature selection on traditional hyperspectral datasets (airborne and satellite).

Assuming there are $p$ classes present over an area in which the samples were collected, the endmember matrix can be written as $S = [s_1, s_2, \ldots, s_p]$. According to Yang et al. [19], with linear mixing of the endmembers, the pixel $r$ can be expressed as in Equation (2):

$$r = S\alpha + n \tag{2}$$

where $\alpha = (a_1, a_2, \ldots, a_p)^T$ is the abundance vector and $n$ is the uncorrelated noise with $E(n) = 0$ and $Cov(n) = \sigma^2 I$ ($I$ is an identity matrix).

Usually, the actual number of classes ($p$) is greater than the known class signatures; i.e., $q < p$. Hence, the uncorrelated noise will have $Cov(n) = \sigma^2 \Sigma$, where $\Sigma$ is the noise

covariance matrix. Therefore, the abundance vector becomes the weighted least square solution, as in Equation (3):

$$\hat{\alpha} = (S^T\Sigma^{-1}S)^{-1}S^T\Sigma^{-1}\mathbf{r} \tag{3}$$

with first-order moment being $E(\hat{\alpha}) = \alpha$ and second-order moment being $Cov(\hat{\alpha}) = \sigma^2(S^T\Sigma^{-1}S)^{-1}$.

The analysis demonstrates that when all the classes are known, the remaining noise can be modelled as independent Gaussian noise. For this application, when meeting such sampling criteria was difficult and there were unknown classes present, noise whitening was applied first. Yang et al. [19] and Su et al. [23] performed the optimal band selection on traditional hyperspectral datasets, and used all the pixels for the background noise ($\Sigma$) estimation. In this case, the background pixels' noise was calculated using background class spectra and bare-earth spectra collected through ground-based sampling. The background plus noise covariance is denoted as $\Sigma_{b+n}$; this estimate was used in this study. The estimate of the unknown class pixels is based on the likelihood of the unknown class (or the class of no interest) being present around the sampled class of interest. In scenes where all endmembers are of known classes (or the target classes of interest), noise estimation $\Sigma_{b+n}$ is not required, which is an unlikely condition in a spectrally complex swamp environment [7].

The identified optimal bands should allow minimal deviations of $\hat{\alpha}$ from actual $\alpha$ [23]. With the partially known classes, the criterion function is equivalent to minimizing the trace of the covariance, as in Equation (4):

$$\underset{\Phi^S}{\arg\min}\{trace[(S^T\Sigma_{b+n}{}^{-1}S)^{-1}]\} \tag{4}$$

where $\Phi^S$ is the selected band subset. The resulting band selection algorithm is referred to as the MEAC method [23].

The optimizer returns a suitably identified set of wavelength bands with the lowest cost criterion values (Equation (4)), upon successful completion of the PSO-MEAC algorithmic iterations (Figure 2).

### 2.5. UAV-Hyperspectral Survey and Assessment

After the identification of a set of optimal bands through the *data-driven* PSO-MEAC approach, the FPI hyperspectral sensor was programmed to acquire using the suitable narrow wavelength bands. A UAV-hyperspectral mission was carried out in pre-planned waypoint acquisition mode with 85% forwards and 75% lateral overlap from a flying altitude of 50 m. The sensor exposure time was set at 10 ms per band to provide good radiometric image quality for the existing illumination conditions. The UAV-hyperspectral survey was performed around two hours of local solar noon and in clear weather conditions with no clouds. This was done to avoid both the effect of significant illumination variations and shadows cast by clouds during the aerial image acquisition. However, due to the experimental site being situated in a low latitudinal region in the southern hemisphere (34°21′24.0″S, 150°51′51″E) with the sun projecting a shallow incidence angle, the issues of the shadows projected by trees and other tall vegetation was unavoidable. In addition to the *data-driven* PSO-MEAC tuned survey, another aerial survey was performed with an *indice-based* [7] wavelength selection approach, using the same UAV flight characteristic and sensor exposure configuration. A band stabilization workflow was adopted to co-register spatial shifts between bands in hypercubes, from both the aerial acquisition modes [26]. Further, the regular radiometric, illumination adjustment, mosaicking, and geometric correction procedures for hypercubes were carried out [7]. The UAV-hyperspectral orthomosaics achieved a high spatial resolution of 2 cm in ground sampling distance.

A supervised support vector machine (SVM) classifier was used to classify the hyperspectral datasets into constituent classes. The SVM is an efficient kernel-based machine learning classifier suitable for high-dimensional feature spaces, which is well used in classifying hyperspectral datasets [32–34]. The classification was performed as an evaluation

step to compare the efficacy of wavelengths identified through *data-driven* PSO-MEAC and *indices-based* approaches. As the fundamental objective in this study was to simply evaluate the two methods, and not to achieve superior accuracies in classification, involving complex classification algorithms were deemed needless. Standard parameter settings—a radial basis function with a kernel gamma function of 0.167, a penalty parameter of 100, and a pyramid level of 5—were used for the SVM classification. The overall and individual class classification accuracies were computed using the ground truth training samples.

For evaluating the efficacy of PSO-MEAC-identified bands through classification, a total of 120 ground truth measurements were collected for shrub-type swamp vegetation through a rigorous field survey, and 120 ground truth polygons were identified through visual interpretation of high-resolution hyperspectral data. The sampled ground-based (120) and image-based (120) polygons were randomly divided into 1:1 mutually exclusive sets of training and test samples, i.e., 60 ground and 60 image-based polygons for each training and test group. The ground truth training set was used to train the SVM classifier, and the test samples were used to compute the overall accuracy (OA), kappa ($\kappa$), and confusion matrix to evaluate the classification accuracies. The spectral data from training and test sample polygons were obtained from the UAV-hyperspectral datasets in corresponding *data-driven* PSO-MEAC and *indices-based* modes.

## 3. Results and Discussion

This section details the results and discussion of optimal band selection using *data-driven* PSO-MEAC workflow, and its evaluation against the *indices-based* approach.

### 3.1. Optimal Band Identification Using PSO-MEAC

The PSO-based optimal band identification workflow determines a list of suitable bands according to the MEAC cost criterion. The PSO-MEAC workflow was executed with a population size of 100, an inertial weight of 0.98, and a maximum number of iterations of 500. A total of 15 bands, i.e., $k = 15$, were identified, based on the maximum band capacity of the FPI sensor for on-board UAV data acquisition mode in an un-binned setting ($1010 \times 1010$ pixels).

The selected combination of bands gets re-configured at every iteration to minimize the cost function (Figure 2). A new combination of bands is designated optimal if the combination achieves the best (or minimum) cost. To analyze the performance of the in-field optimal band identification and sensor tuning using the PSO-MEAC approach, a set of internally computed parameters (criterion cost and index of runs) were logged at every iteration (Figure 3). The PSO-MEAC approach determines the suitable combination of bands (or band-index) using the cost criterion (Equation (4)). The reduction of the best cost value signifies the learning curve for the optimization workflow (Figure 3a). At every iteration, the cost associated with the previous band-index is compared with the new band-index. A record of these parameters reveals the process of convergence to the desired solution by the implemented metaheuristic workflow. A measure of final cost and plot of identified optimal band combination is also produced. It can be seen that using the PSO-MEAC method, better (i.e., smaller) values of cost criterion can be achieved. Each iteration may produce slightly different band combinations according to the cost criterion, as shown by the plot of the index of runs in (Figure 3b). The final cost of the PSO-MEAC was $-7.7 \times 10^{-9}$. At this stage, the identified band indices were 56, 88, 101, 119, 151, 172, 211, 217, 251, 284, 303, 326, 341, 360, and 380 (Figure 3c). The corresponding FPI wavelengths were 555.33, 587.21, 600.34, 618.21, 650.39, 671.02, 710.12, 716.11, 750.19, 783.46, 802.35, 825.28, 840.15, 859.53, and 880.43 nm with respective FWHMs of 9.81, 10.62, 9.88, 12.17, 10.78, 11.77, 9.78, 9.61, 9.58, 10.60, 10.56, 10.49, 13.69, 13.12, and 13.27 nm.

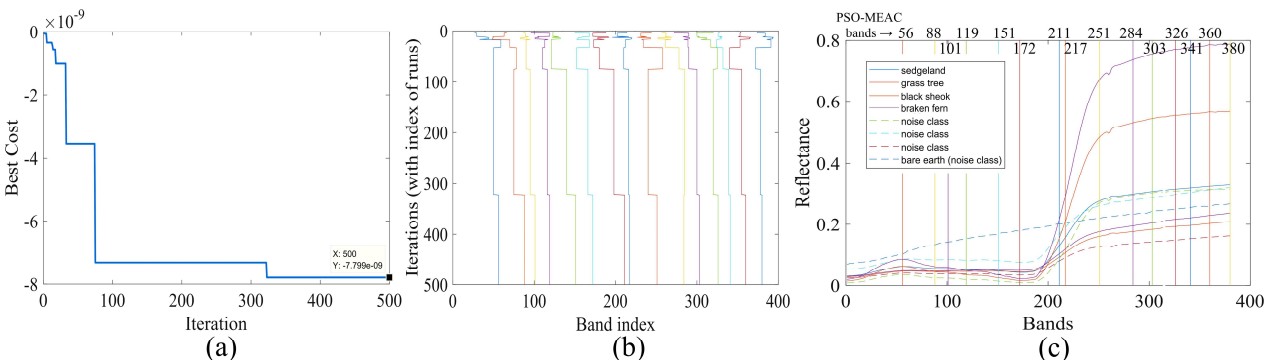

**Figure 3.** Optimal band selection: (**a**) a plot demonstrating variation of cost criterion with the PSO-MEAC iterations, (**b**) bands selected in each iteration, and (**c**) a plot of the identified optimal bands overlaid on the class spectra. The cost criterion was progressively minimized with the number of iterations. The variations of band position with the index of runs in every iteration provide insights into the functioning of PSO-MEAC. Overall, the PSO-MEAC-identified bands are well distributed over the key wavebands with maximal variation between inter-class reflectance.

The PSO-MEAC workflow uses a complex high-dimensional search strategy, producing several intermediate local and global combinations of bands, so the final solution may not be the same with every execution. Previous implementations of PSO-MEAC [23] focused on minimising the number of bands in optimal configurations, which is suitable for dimensionality reduction techniques in traditional airborne or satellite hyperspectral imaging, with a complete set of bands already acquired. In the proposed method, the number of bands to be identified is predefined by the user, which makes it important to use the FPI sensor to its fullest potential (i.e., hypercube band capacity at desired spectral binning) to acquire the maximum possible information in the optimal configuration. To evaluate the computational complexity, the PSO-MEAC workflow was programmed in MATLAB (ver. 9.5) and implemented as a GUI module to run on a portable field data acquisition computer with 1.5 GHz processor and 512 MB memory. The module took roughly 4 to 5 min for every 500 iterations with the selected number of class samples. This demonstrates the operational efficiency of the system, despite having a complex search hierarchy, and it is usable for pre-tuning the programmable FPI sensor in a UAV-hyperspectral survey for optimised wavelength selection.

Acquisition and identification of optimal bands using characteristic spectral signatures of individual swamp species have been traditionally performed using the separability of the spectrum at respective wavelength bands. In this study, the employed PSO-MEAC-based search strategy automatically analyses and identifies wavelength bands based on maximum separability of the reflectance using the MEAC cost criterion function. The field spectrum collected for each shrub-type vegetation species is shown in (Figure 3c), and the identified wavelength band positions are shown using a set of superimposed vertical lines. Our approach has been implemented using a GUI-based interface on a portable data acquisition computer, which enabled rapid analysis of spectral signatures and identification of suitable wavelength bands. The developed technique and tools were found to be efficient in a field environment during surveying.

### 3.2. Classification

The comparative evaluation between the *data-driven* PSO-MEAC and *indices-based* wavelength tuning approaches was performed using an SVM classifier. Two dedicated datasets (*data-driven* PSO-MEAC and *indices-based*) were collected from the swamp. The scene was primarily comprised of three shrub-type vegetation classes (i.e., grass trees, pouched coral ferns, and Sedgeland complex) and two tree-type vegetation classes (i.e., black sheoak and eucalyptus). A small portion of the area was bare of vegetation cover and was treated as a separate "bare earth" class. Therefore, a total of six classes were used in the classification-based comparative evaluation. The optimal bands identified using the

*data-driven* PSO-MEAC approach produced better results compared to the *indices-based* approach, with the SVM classifier. Combining the optimal bands identified using the *data-driven* PSO-MEAC with the SVM classifier produced an overall accuracy of 85.16% and a kappa coefficient of 0.73, whereas the *indices-based* approach produced an overall accuracy of 76.54% and a kappa coefficient of 0.67. The comparative classification maps for the *indices-based* PSO-MEAC and *data-driven* approaches produced using the SVM classifier are shown in Figure 4.

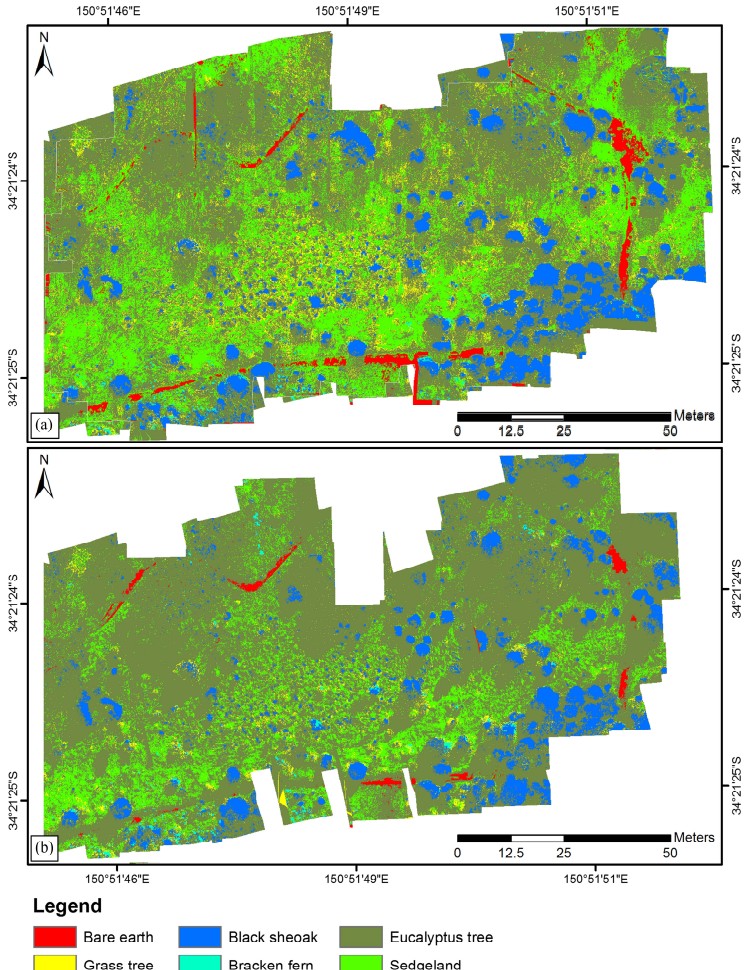

**Figure 4.** Classification map of the swamp site's vegetation classes and species produced using a support vector machine classifier with (**a**) *data-driven* (PSO-MEAC) optimal band identification and (**b**) *indices-based* band selection.

The producer's accuracy or error-of-omission refers to the conditional probability that certain land-cover of an area on the ground is correctly mapped, whereas the user's accuracy or error-of-commission refers to the conditional probability that a pixel labeled as a certain land-cover class in the map belongs to that class [35]. The producer's and user's accuracy for each class with the best classification method, *data-driven* PSO-MEAC, are shown in Table 1. With the exception of the "grass tree" class, overall the accuracy for each class was satisfactory (>70%), particularly when differentiating between swamp-type (Sedgeland complex) and non-swamp-type (*Eucalyptus*) vegetation. The results also indicate the potential of the process for distinguishing certain critical non-swamp-type terrestrial species (black sheoak and bracken fern) within the swamp environment. Increases in the proportions of these terrestrial species in a swamp indicate changes in the swamp hydrology. No changes in the proportions of terrestrial species (or changes within equilibrium limits) indicates the stability of hydrology and peat moisture levels. These

results, therefore, demonstrate the usefulness of the method for directly mapping the changes induced in a swamp environment due to the fluctuation of groundwater level.

**Table 1.** Evaluation of classification accuracy achieved using the *data-driven* (PSO-MEAC) method against *indices based* band selection.

| Class | *Data Driven* (PSO-MEAC) | | *Indices Based* | |
|---|---|---|---|---|
| | Producer's Accuracy (%) | User's Accuracy (%) | Producer's Accuracy (%) | User's Accuracy (%) |
| Bare earth | 91.57 | 98.52 | 22.27 | 89.43 |
| Grass tree | 77.00 | 73.54 | 5.92 | 49.45 |
| Black sheoak | 97.33 | 83.20 | 94.28 | 78.93 |
| Bracken fern | 71.43 | 78.44 | 13.08 | 70.83 |
| Eucalyptus tree | 81.55 | 81.13 | 89.10 | 71.16 |
| Sedgeland complex | 88.28 | 80.35 | 41.42 | 61.64 |

## 4. Conclusions

Identification of optimal bands for vegetation monitoring has been an ongoing research problem in hyperspectral remote sensing. The issue is significant in a spectrally complex environment with diversity in vegetation species, such as swamps and wetlands. Extensive surveys and post-processing solutions have been recurrently used in different swamp-type environments. The study presents an innovative approach for in-field rapid identification of spectrally significant wavelength bands. The developed method was employed to tune a programmable hyperspectral sensor before UAV borne surveys. The method was implemented through a metaheuristic workflow based on particle swarm optimization (PSO), with minimum estimated abundance covariance (MEAC) as the cost selection criterion function. A portable in-field hyperspectral signature collection system was devised using the tuneable FPI hyperspectral sensor. The set-up improved the collection of class spectra and background noise spectra, which were then used to identify the optimal band configuration. The method identifies the optimal bands based on representative class spectral signatures, avoiding the requirement of extensive in-field sampling. Additionally, the method works perfectly when the number of sample observations is less than the total number of potential hyperspectral bands, which is not possible with other dimensionality reduction methods, such as PCA. The method was successfully tested to identify a set of optimal bands for maximizing the spectral differentiation of swamp-type vegetation species and communities. The algorithm could be tuned to robustly incorporate vegetation trait retrieval by changing the criterion function. The approach would be valuable to environmental mapping, precision agriculture, phenotyping, and forestry to estimate qualitative phenotypic traits such as chlorophyll content, photosynthetic capacity, and biomass; and for studying vegetation under different treatments or biotic and abiotic stresses.

**Author Contributions:** B.P.B. and S.R. conceived of the experiment. B.P.B. conducted the experiments, data analysis, and writing of the original draft. S.R. conducted project administration, manuscript review, and editing. All authors have read and agreed to the published version of the manuscript.

**Funding:** This research received no external funding.

**Institutional Review Board Statement:** Not applicable.

**Informed Consent Statement:** Not applicable.

**Data Availability Statement:** Data are not available due to non-disclosure agreements.

**Conflicts of Interest:** The authors declare no conflict of interest.

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
