# Peer review of "A Particle Swarm Optimization Based Approach to Pre-tune Programmable Hyperspectral Sensors"

_remotesensing, doi:10.3390/rs13163295_

Round 1

Reviewer 1 Report

The scope seems adequate for the category of a technical note. The literature review, the materials and methods are well-written. Results and discussion need improvements (see below).

Line 93: Please briefly describe 1PSO and 2PSO used in the reference.

Line 175: Figure 1 is useful for understanding the procedure for generating ground-truth data; are the scenes obtained in (c) continuous? If not, what was the interval used?  

Line 276: The "ref" within the parentheses should be specified.

Line 284: The explanation of polygons is somewhat ambiguous. Does a polygon mean each snapshot obtained using the method described in Fig. 1?

Line 364: The accuracy of each class should be provided at a minimum along with the overall accuracies of data-driven and indices-driven approaches. It is also desired to provide the computed confusion matrix if available.

Line 374: What do the producer and the user exactly mean? Are they defined with respect to the GUI mentioned in the previous paragraph?

Reviewer 2 Report

This study presents a method to identify the optimal bands for vegetation monitoring purposes. An innovative approach was developed for in-field rapid identification of spectrally significant wavelength bands for a given environment to program tunable hyperspectral sensor acquisition before UAV borne surveys. The method was implemented using a PSO with minimum estimated abundance co-variance (MEAC) as the cost selection criterion function. Finally, a portable in-field hyperspectral signature collection system was devised using the tunable FPI hyperspectral sensor.

Generally, the article is well written but there is no real problem contextualization. I would include in the introduction a short note on precision agriculture, remote sensing and the use of drones equipped with hyperspectral image analysis sensors, used for these purposes, to better contextualize the study.

In addition, the aim is confused: it is more of an activities list.

I will consider the work for a publication only if a thorough review will be done.

Minor comments

Not all acronyms are spliced (they should be deleted from the abstract). Report either an initial table with all the explanations acronyms or always specify them before writing.

Do not use pronouns in the text (e.g., we).

Generally, figure’s captions are not very detailed and explanatory.

Major comments

Introduction

Include in the introduction a short note on precision agriculture, remote sensing and the use of drones equipped with hyperspectral image analysis sensors, used for these purposes, to better contextualize the study.

L.75: various works related to data driven strategies are rightly listed but their results are not reported (i.e., how effective these methods are for the reported purpose). Please insert.

There is not a real aim but a list of activities that have been carried out. Insert it to make the activity more understandable.

Materials and methods

Paragraph 2.1: enter the GPS coordinates of the study areas.

Introduce a paragraph on the setup of the management of climatic conditions limiting the acquisition of images (e.g., shadows, clouds, etc.).

  1. 197: insert version of MATLAB.

Results

Insert tables to better understand the obtained results.

Conclusions

This section is not well organized.

Round 2

Reviewer 1 Report

Thank you for your response. Hope that the method will promote environmental sustainability.

Reviewer 2 Report

The manuscript in the present form can be published in Remote Sensing as the authors have been improved it. Parts that better contextualize the work have been added (e.g., precision agriculture, remote sensing and the use of drones).

I don’t need to review another version because I accept the work in the present form.